# Nonuniform Bessel-Based Radiation Distributions on A Spherically Curved Boundary for Modeling the Acoustic Field of Focused Ultrasound Transducers

**Mario Ibrahin Gutierrez** [1] **, Antonio Ramos** [2] **, Josefina Gutierrez** [3,*] **, Arturo Vera** [4] **and Lorenzo Leija** [4]

1 CONACYT—Instituto Nacional de Rehabilitación, Subdirección de Investigación Biotecnológica, División de Investigación en Ingeniería Médica, Calz. Mexico-Xochimilco 289, Tlalpan, Mexico City 14389, Mexico; m.ibrahin.gutierrez@gmail.com

2 Consejo Superior de Investigaciones Científicas, CSIC, Instituto de Tecnologías Físicas y de la Información, R&D Group. Ultrasonic Signals, Systems and Technologies, C/Serrano 144, 28006 Madrid, Spain; aramos@ia.cetef.csic.es

3 Instituto Nacional de Rehabilitación, Subdirección de Investigación Biotecnológica, División de Investigación en Ingeniería Médica, Calz. Mexico-Xochimilco 289, Tlalpan, Mexico City 14389, Mexico

4 Centro de Investigación y de Estudios Avanzados del IPN, Cinvestav-IPN, Department of Electrical Engineering, Bioelectronics Section, Av. IPN 2508, Gustavo A. Madero, Mexico City 07360, Mexico; arvera@cinvestav.mx (A.V.); lleija@cinvestav.mx (L.L.)

\* Correspondence: jgutierrez@inr.gob.mx; Tel.: +52-555-999-1000

**Abstract:** Therapeutic focused ultrasound is a technique that can be used with different intensities depending on the application. For instance, low intensities are required in nonthermal therapies, such as drug delivering, gene therapy, etc.; high intensity ultrasound is used for either thermal therapy or instantaneous tissue destruction, for example, in oncologic therapy with hyperthermia and tumor ablation. When an adequate therapy planning is desired, the acoustic field models of curve radiators should be improved in terms of simplicity and congruence at the prefocal zone. Traditional ideal models using uniform vibration distributions usually do not produce adequate results for clamped unbacked curved radiators. In this paper, it is proposed the use of a Bessel-based nonuniform radiation distribution at the surface of a curved radiator to model the field produced by real focused transducers. This proposal is based on the observed complex vibration of curved transducers modified by Lamb waves, which have a non-negligible effect in the acoustic field. The use of Bessel-based functions to approximate the measured vibration instead of using plain measurements simplifies the rationale and expands the applicability of this modeling approach, for example, when the determination of the effects of ultrasound in tissues is required.

**Keywords:** focused transducer; acoustic field; nonuniform radiation distribution; Bessel radiation distribution; spherically curved uniform radiator; rim radiation; Lamb waves; finite element modeling

---

## 1. Introduction

In recent years, the use of focused ultrasounds (FUS) has been increased in biological applications for both high intensity and low intensity modalities [1–5]. A high-intensity focused ultrasound is used for the rapid destruction of tissues by thermal ablation [2,3,6], for example in oncology, while low-intensity applications are based on producing midterm hyperthermia and nonthermal ultrasonic therapy [3,7], with multiple possible applications [4,8,9]. Among the non-ablating FUS applications

reported in literature can be mentioned the low-intensity pulsed ultrasound (LIPUS) using focused transducers [8], drug delivery in deep tissues (as blood–brain barrier disruption) [10], gene transfer therapy [11], and sonothrombolysis [12,13], among others. In all these applications, the control of the dose in the target volume should be precise during long periods of time to avoid cell death in non-treated zones [12,14]. Noninvasive (and non-expensive) technologies to monitor the temperature in the treated zone [12], more precise and simpler calibration techniques for FUS transducers to determine effective radiating parameters [15,16], in conjunction with accurate and simple computational models capable of effectively representing the acoustic fields of real FUS transducers are required. This could provide more information to adequately study the produced effects along the ultrasound pathway and to quantify undesired consequences in surrounding tissues, previous the application of the therapy. However, in medical applications, the use of simple, but inefficient, ideal models for the acoustic field of FUS transducers is a common and not very questioned practice [17–19].

One of the first approaches to calculate the acoustic field of curved radiators is the O'Neil solution proposed in 1949 [20]. This solution is based on the Rayleigh integral, and it assumes a spherically curved uniform radiator (SCUR) oscillating with a uniform velocity distribution. Although this approximation could be appropriate for many modern transducers at the focus, in the regions where the pressure amplitudes are highly affected by diffraction, e.g., before the focus, the discrepancies are very evident [17]. These variations between "ideal" theory and experiments occur because the assumption of a normal uniform velocity distribution is very conservative and, usually, unreal [17,21,22]. The rather complex vibration of piezoelectric plates not only is composed by a thickness-extensional (TE) vibration mode but also includes contributions of radial modes [23,24], edge waves [17], and Lamb waves [21,25], whose effects are more noticeable under a continuous regime [17]; this vibration can be more complicated if we consider that the piezoelectric plate is not vibrating freely, but it is somehow clamped by its edge producing a higher vibration amplitude at the center of the plate with an attenuated vibration at the edge [26,27]. This complex vibration occurs in both planar and concave plates, and it should be accounted for when producing models of acoustic fields for more accurate results.

In this paper, we are proposing an approach to model the acoustic field produced by FUS transducers in a low-intensity regime (considering linear propagation) using polynomial-Bessel based functions as nonuniform radiating distributions on a curved surface. The reason of proposing these functions is based on the reported vibration patterns produced in piezoelectric curved disks composed of a main vibrating thickness-extensional (TE) mode generating a wave in the thickness direction and a second component of Lamb waves propagating radially [17,28,29]. These two components are modified when the disk is fixed to the transducer case, which produced a combined vibration pattern that can be approximated by a polynomial-Bessel based function, in accordance with the measurements for any specific transducer. With this approach, we got better results than the widely used SCUR, which can be comparable with the results using the intuitive approach for modeling the acoustic field using the velocity distribution measured on the radiating surface [29,30]. Using analytical functions instead of measurements will permit to propose mathematical algorithms to produce realistic acoustic fields of actual FUS transducers; however, these demonstrations are beyond the scope of this paper. The future applications of these new models are open, since the assumptions taken for this work are not particular.

## 2. Materials and Methods

In this paper, the acoustic field of a FUS transducer has been modeled. A 2 MHz spherically focused transducer with a 20 mm nominal focal length and a 20 mm of nominal aperture (Onda Corporation, Sunnyvale, CA, USA) was used for the experiments; these values usually differ from the measured ones as it will be discussed later [15]. This kind of transducer, designed for high-intensity applications but used here for low-intensity measurements, has its negative terminal exposed, i.e., it does not have a nonconductive front-layer. This aspect will be useful for the explained below curvature measurements.

### 2.1. Measurement Setup

Three different sets of measurements were carried out for this work using the setup shown in Figure 1. For the first set, the curvature of the FUS transducer was determined by taking advantage of its exposed negative terminal on the radiating surface; for this, a multimeter (Fluke 289, Fluke Corporation, Everett, WA, USA) and a 3D positioning system (Onda Corporation, Sunnyvale, CA, USA) driven in manual mode were used. The transducer was placed at the final position for acoustic field measurements (to be made after) in which the transducer will emit in positive $z$-direction; this first measurement was made in the air. Then, the multimeter was set to measure electric continuity, and one of its probes was attached to the positioning system while the other was connected to the negative terminal of the transducer. The measurement probe was moved step-by-step (step resolution of 6.36 $\mu$m) towards the transducer surface, down in the $z$-direction (relative to the direction of the ultrasound emission), for a fixed $x$-coordinate until the probe touched the transducer conductive surface, indicated by a "beep" from the multimeter. This procedure was repeated diametrically for every $x$-coordinate from $-14.573$ mm to 14.573 mm with a step resolution of 0.3312 mm, covering the transducer case and the curved piezoelectric plate. The used step-resolution was smaller than a half of a wavelength of 2 MHz ultrasound in water at 20 °C (approx. 0.38 mm). The determined $z$-coordinates for all the "$x$" were saved for further use. These final coordinates were used for the second below explained measurement.

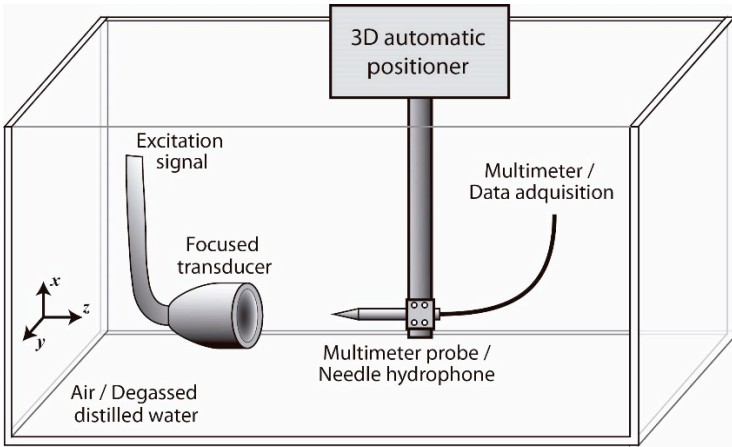

**Figure 1.** The setup for the curvature/acoustic pressure measurements.

For the second set, the transducer was immersed in a tank filled with degassed distilled water carefully keeping intact the transducer setup used for the first set (see Figure 1). This second set of measurements was carried out using the previous obtained coordinates for the curvature to determine the radiation pattern "very close to the transducer surface". The radiation measured at this region would closely represent the vibration distribution of the curved piezoelectric disk, assuming the effect of other regions of the radiating surface is negligible. For this, a wideband needle hydrophone PZTZ44-0400 (Onda Corporation, Sunnyvale, CA, USA) with a 40 $\mu$m aperture and sensitivity of $-260$ dB referred to 1 V/$\mu$Pa was mounted on the positioning system, replacing the multimeter probe. The measurements were made at the positions determined previously but 1 mm in front of the transducer surface (in $z$-direction). The transducer was driven with a wave generator (Array 3400, Array Electronic Co., Taiwan, China) using s 20 Vpp sine tone-burst; the received data were recorded in a PC. The ultrasound signals were recorded for 1, 5, 10, and 20 sine cycles to determine the radiation pattern under different excitation conditions but, more specifically, to know in which number of cycles the vibration presents a quasi-stationary pattern. The data were post-processed in MATLAB (R2017a, MathWorks, Natick, MA, USA) to determine the peak-to-peak values at each spatial point and the full radiation distribution for each excitation condition.

The third set of measurements was performed to determine the full acoustic fields of the focused transducer. This was measured using the previously described setup. The acoustic fields were obtained by saving only the peak-to-peak acoustic pressure at each point in the radiated volume. It was recorded on an XZ plane covering the transducer dimensions starting at $z = 2$ mm and finishing at $z = 50$ mm from the transducer case, with a step resolution of 0.3312 mm in the *x*-direction and 0.5000 mm in the *z*-direction. The XY planes were measured at two depths, 0.2 cm and 1.7 cm (at the focus), using a step resolution of 0.3312 mm in both directions; these planes were obtained to verify the symmetry of the acoustic radiation. As formerly, the acoustic fields were captured for 1, 5, 10, and 20 sine cycles to determine the conditions for a nearly continuous emission pattern. The data were postprocessed in MATLAB to reconstruct the fields.

### 2.2. Acoustic Field Modeling Using FEM

The acoustic field was modeled using the finite element method (FEM) based on the geometry shown in Figure 2. The software used for the FEM processing was COMSOL Multiphysics (COMSOL AB, Stockholm, Sweden) working in a PC of 8-core 3 GHz microprocessor and 64 RAM (Dell, Round Rock, TX, USA). Based on the cylindrical symmetry of the transducer, the problem was assumed axisymmetric. The validity of this assumption was verified with acoustic field measurements in which the profile of the radiation along the azimuthal coordinate is similar for any angle (data not shown). The mesh in the rectangular part of Figure 2 consisted of 10 square elements per wavelength, i.e., more than 530,000 elements; triangular elements were used only for the zone created with the boundaries 1 and 5 to simplify meshing. The requirements to mesh using squares are very specific, and this kind of element cannot be used in certain curved geometries; triangular elements are sometimes the only option for these complicated zones. The mesh convergence was verified by increasing the mesh resolution, which indicated an error of 0.01% at the focus amplitude between meshing using 9 and 10 elements per wavelength. Using square elements instead of triangular permitted to increase the spatial resolution with the same interpolation functions because this kind of mesh has a larger number of nodes (and therefore more degrees of freedom) than the triangular one with the same number of elements (more nodes per element area); this can be noticed with the reduction of the solution time compared to the time required for solving the problem using the same number of triangular elements. The largest solution time registered for our main model, which was barely the most demanding of computational resources, was 45 s.

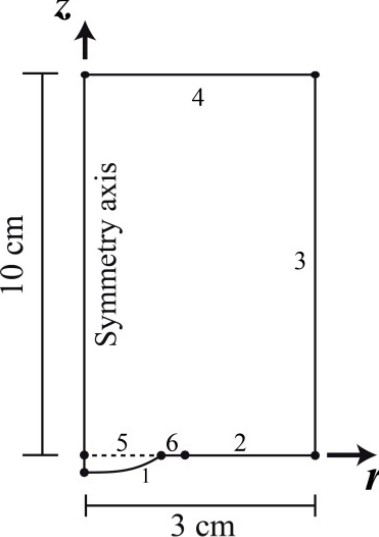

**Figure 2.** The finite element method (FEM) axisymmetric geometry for modeling the focused acoustic field.

It was assumed the linear ultrasound propagation (i.e., rather low acoustic intensity) was in an attenuation-free homogeneous media (degassed distilled water). The ultrasound propagation is determined with FEM based on the homogeneous Helmholtz wave equation for harmonic radiation, assuming a purely harmonic source and a no-frequency dispersion. This equation can be written as

$$\nabla^2 p + k^2 p = 0 \tag{1}$$

where the wavenumber is $k = \omega/c$, $\omega$ is the angular frequency (rad), and $c$ is the speed of sound in water (m/s) assumed constant. Boundary 1 was set with the harmonic normal acceleration. The amplitude of this normal acceleration was adjusted for the different radiation conditions presented in this paper. For instance, in the uniform radiation model of the next section, the amplitude of the harmonic acceleration was set constant along the radius. Usually, the radiation of a boundary is expressed in terms of particle velocity [20,26,31]. In harmonic conditions [32], the particle acceleration $a_0$ on the radiator surface can be obtained by time-deriving the constant-amplitude particle velocity $v_0 e^{j\omega t}$ as

$$a_0 = \frac{d}{dt}\left(v_0 e^{j\omega t}\right) = v_0 \omega e^{j(\omega t + \frac{\pi}{2})} = \frac{\omega}{\rho c} p_0 e^{j(\omega t + \frac{\pi}{2})}, \tag{2}$$

where $\rho$ is the medium density (kg/cm$^3$). The last term was determined by considering $p_0 = \rho c v_0$, which is true in the field very close to the radiator surface [27]. The term $\frac{\pi}{2}$ in Equation (2) is a time phase shift between the velocity and the acceleration, and this is not related to the spatial profile along the transducer surface [32]. Then, under harmonic simulations, using either acceleration or velocity for the radiation distributions of the plate is not relevant if adequate amplitude considerations are taken.

Boundaries 2–4 were configured to match the acoustic impedance of water and to reduce the ultrasound reflections at the walls [27]. Then, $Z = \rho c = 1.5$ MRayls at 25 °C, where $Z$ is the acoustic impedance (MRayls) given by the product of the media density and the speed of sound; the walls were considered to be perfectly flat. However, the dimensions of the FEM geometry (10 cm × 3 cm) were big enough to not have residual reflected waves affecting the region of interest for this application, i.e., a region of 5.0 cm depth and 1.5 cm width after the transducer. Boundary 5 was set with a continuity condition, and it was used only to simplify the meshing. Boundary 6 represents the transducer rim, which was set, accordingly to the measurements, to uniformly radiate a relative pressure of 8% of the average pressure radiated by the curved surface in the effective radiating area; this area was determined as the area producing 95% of the transducer's radiation, as defined in other applications for planar radiators [27]. The use of this effective area instead of the nominal area improved the model results as it will be explained later; this was already proposed by other means in Reference [15]. The rim radiation was included because it represents an important contribution in the field for this transducer, more evident in the post-focus field but with some little effects in the pressure amplitude on the propagation axis (along *z*-axis) in the prefocus zone.

### 2.2.1. Radiator with Uniform Vibration Distribution

Conventionally, the acoustic field produced by planar ultrasound transducers has been represented as the product of the radiation coming from a uniform vibrating surface (the piston in a Baffle and the Rayleigh equation) [33]. This assumption produces adequate results for wideband transducers, in which the vibration is usually damped by the backing material and, thus, produces quasi-uniform displacements along the transducer radiating surface [34–36]. However, narrowband transducers often do not have a backing material (air-backed) which makes the vibration of their piezoelectric components less uniform [27]. These nonuniformities have an important effect in the field.

The acoustic field of focused transducers has been historically modeled following the same supposition of planar plates, in which it is assumed the plate is a slightly curved uniform radiator (SCUR) [20]. For this, the curvature of the piezoelectric plate of the transducer is usually assumed spherical; conversely, this curvature is not often reported in datasheets. The most well-known

theoretical proposal to determine the acoustic field produced by curved surfaces was made by O'Neil in 1949 [20]. This model works adequately for low-power wideband transducers with a backing material, and it has produced poor results when used to model the acoustic field of air-backed narrowband radiators [17,36]. In order to compare our proposal with the most used ideal approach of curved transducers, the acoustic field produced by a SCUR was determined. For this, a constant amplitude harmonic normal acceleration was used in the curved boundary 1, in Figure 2.

### 2.2.2. Radiator with "Classical" Nonuniform Vibration

Nonuniform vibration distributions have been proposed to generate consistent acoustic field models of some planar radiators but with a limited range of applicability. These are based on the assumption that, under certain conditions, a radiator can behave not only as a piston but also as a membrane and a clamped plate [26,31,37]. The general equation of "classical" nonuniform radiation distributions include the two more common theoretical conditions for fixing the edge of a rigid piezoelectric plate [38]: 1) a plate with simply supported edges that restrict edge movement in any direction but allow rotation by the edge (simply supported radiator) and 2) a plate with clamped edges that restricts the movement and rotation in any direction (clamped radiator). The general equation for the nonuniform radial acceleration $a_0(r)$ of the surface of a plate radiator can be expressed by [26,31,37]

$$a_0(r) = \sum_{mn} A_{mn} \left[ 1 - \left( \frac{r}{R} \right)^{2m} \right]^{n+1}, \tag{3}$$

where $R$ is the fixed radius of the plate (m), $A_{mn}$ is the adjustable amplitude of the acceleration, and constants $m$ and $n$ will determine the shape of the distribution. However, only two cases have been reported to represent an actual meaning in Equation (3), specifically, when $n = 0$, the radiation distribution corresponds to the simply supported radiator (SSR) for $m = 1, 2, 3, \ldots$; when $m = 1$, the radiation distribution is that of a clamped radiator (CR) for $n > 0$. This radial acceleration is set in boundary 1 shown in Figure 2. The summation in Equation (3) can be reduced after assuming that the first vibration mode dominates [31].

### 2.2.3. Radiator with Bessel Acceleration

The vibration distribution of a piezoelectric plate is closer to a combination of Bessels than a uniform distribution [38–40]. This behavior depends on the material's piezoelectric properties, the constraint conditions, the coupling layers, and the shape. When the plate is concave, the vibration of the concave-shape piezoelectric plates can also be composed of Bessel-like vibration distributions due to the Lamb waves [17]. The components affecting the vibration of an ultrasound transducer are vast and difficult to quantify. Then, for the proposal of this paper, the vibration was determined by measuring the acoustic pressure very close to the radiating surface [27]. If we suppose the ultrasound behaves as a plane wave in the region very close to the radiating surface (when $r \to 0$), the acoustic pressure is related with the acceleration as shown in Equation (2).

Thus, the proposal of this paper is to use, as the nonuniform radiation distribution, a Bessel-based function composed by two sub-functions [27]. For this, the acceleration on the curved radiating surface will be expressed by

$$a_0(r) = A_0 f(r) \cdot g(r), \tag{4}$$

where $r$ is the variable radius related to each point in the curved boundary and $A_0$ is the amplitude of the emitted acoustic pressure, provided that the amplitude of the product of $f(r)$ and $g(r)$ is 1 at $r = 0$. The function $f(r)$ is given by the acceleration of Equation (3), and it represents the effect of attaching the piezoelectric plate to the transducer case (edge clamping), then $f(r) = \sum_{mn} A_{mn} \left[ 1 - (r/R)^{2m} \right]^{n+1}$. This function can be determined by adjusting the parameters to get a correct representation of the measured profile, which can be evaluated by correlation.

The function $g(r)$ represents the radial measured peak distribution (MPD) shown in Figure 3. This was approximated using the Bessel-based function

$$g(r) = C_1 J_0\left(\beta_{2N}\frac{r}{R}\right) + C_2. \tag{5}$$

Equation (5) was proposed after considering the reports and simulations about Bessel patterns in the vibration distribution on some transducer surfaces [17,28,29]. For the profile measured for the transducer used for this paper, we used a pure negative Bessel $J_0$ mounted over either an SSR or CR function to approximate the experimental MPD. Then, the Bessel-SSR and Bessel-CR distributions were obtained by combining Equations (5) and (3) into Equation (4), with adequate constants. Here, $C_1$ and $C_2$ are the coefficients that control the amplitude and the offset of Equation (5), respectively, and $\beta_{2N}$ is the $2N$ zero of $J_0$, where $N$ is the number of peaks in the MPD. When $C_1$ is close to zero, the acoustic field is close to the ideal uniform radiation (SCUR) for $C_2 \neq 0$; for practical concerns, the average amplitude into the effective radiating radius of the function $a_0(r)$, after selecting adequate functions $f(r)$ and $g(r)$, should be 1, which can be adjusting with an adequate value of $A_{mn}$. The effective radius was determined by calculating the effective area containing 95% of the total radiation. Then, the amplitude of the emitted pressure can be controlled with $A_0$. In Figure 3, $C_1 = -0.3$, $C_2 = 0.9$, $R = 1$ cm, $m = 1$, $n = 0.3$, $N = 5$, $A_{mn} = 11.7$, and $\beta_{2N} = 30.67$, which is the tenth zero of Bessel $J_0$. The values of $m$ and $n$ of Equation (3) were determined by the values producing the maximum correlation with the measured data. The Bessel-SSR of Figure 3 requires $n = 0$; Bessel-MOD (MOD, modified Bessel-SSR) have the same constants as the Bessel-SSR, but $n = 0.3$. These values are highly dependent on the transducer construction, not only on the piezoelectric constants and disk geometry; they are dependent on the way the disk was attached to the case, the thickness and properties of extra layers (electrodes, glue, etc.), and the mechanical properties of the case. Because of these reasons, it would not be possible to specify any rule to determine the parameters of Equations (3) and (5), neither by simple transducer inspection nor even after knowing the transducers' electrical characteristics. Actual measurements of the emitted field very close to the transducer should be made.

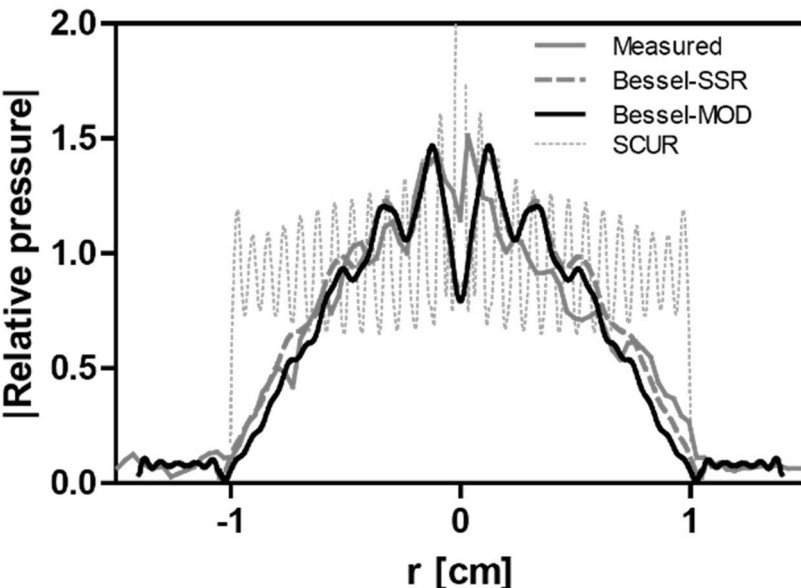

**Figure 3.** The measured acoustic pressure at 1 mm from the radiating concave surface along its radius compared with the uniform and proposed Bessel distributions. The graphs are normalized with the average radiated pressure at the radiator surface.

## 3. Results

In this paper, we propose a Bessel-based nonuniform vibration distribution for the surface of a spherically curved radiator to get a modeled acoustic field closer to a real measured field. Three classical approaches based on uniform (SCUR) and nonuniform distributions are included to contrast our main results. Figure 3 shows the measured pressure profile at 1 mm from the piezoelectric curved plate that is composed of local peaks distributed along the radius. This distribution can be represented with Equation (4), combining an adequate Bessel function (Equation (5)) and a function representing the radiator clamping using the traditional nonuniform approaches (SSR or CR with Equation (3)). Using a radiating function instead of the measured data for determining the acoustic field could permit the generalization of the radiation coming from this kind of transducer and eventually simplify the calculations by proposing analytic solutions for certain conditions. Although SSR and CR nonuniform distributions have been suggested since the 1960s decade for planar radiators, these have not been formally proposed for curved radiators [26,37]; their inclusion in this paper is not just for comparison purposes but as two valid alternatives for producing certain types of acoustic fields. Then, the use of SSR/CR nonuniform distribution for curved radiators is not discarded by this work, so this can be an effective approach if the operation conditions of the transducer produce these kinds of vibration profiles.

In Figure 4, it is shown the relative pressure profile along the propagation axis ($z$-direction) of classical SCUR and two "traditional" nonuniform vibration distributions compared with the measured field. The improvement in the fields using nonuniform distributions on the radiator can be noticed. When the vibration of a curved radiator is uniform, the field has very sharp local peaks along the propagation axis that are smoothed when the emitted radiation is gradually reduced at the plate rim, as in the proposed nonuniform approaches for $r \to R$. The concordances of these fields and the measurements are much better, except in the prefocus zone in which the diffraction patterns differ. In both nonuniform approaches, the post-focus zone is smoother, following the same tendency as the measured profile. There is an after-lobe at about 3 cm, which is well-represented by both nonuniform approaches but only when those included the rim radiation. Without this component, this lobe disappears [41], keeping the most other components of the field. This rim radiation was also included in the main results of this paper.

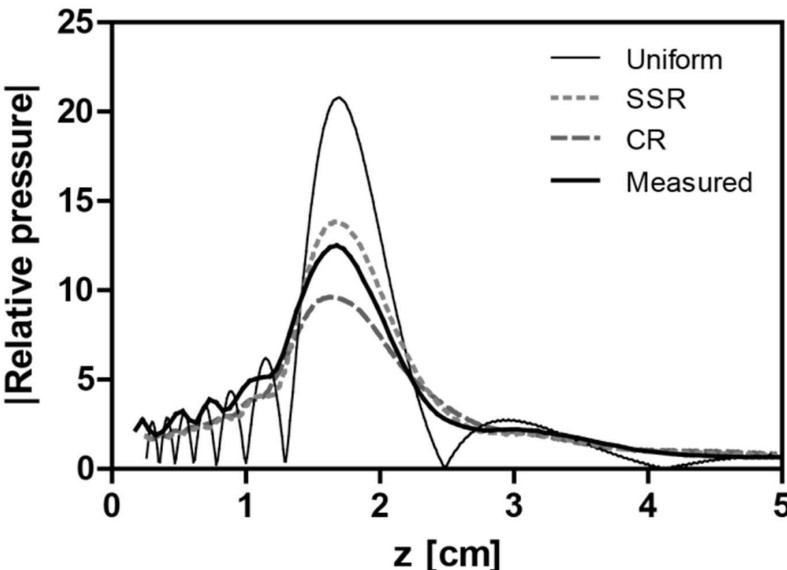

**Figure 4.** The amplitude of the relative acoustic pressure along the propagation axis of the spherically curved radiator with uniform distribution (SCUR), simply supported radiator (SSR), and clamped radiator (CR) conditions vs. measured field. The pressure is relative to the average pressure on the radiator surface of each condition.

The results of using the Bessel-based radiation distributions are shown in Figure 5. For these proposals, Equations (3) to (5) were used to set the acceleration of the curved radiating boundary; also, the radiation from a 3.5 mm rim was included (boundary 6). For the Bessel-SSR approach, it can be noticed that the acoustic field at the prefocal zone has a peak distribution very close to the measured field, and the post-focal zone has the after-lobe produced by the rim radiation with an amplitude equivalent to that of the measurements. The Bessel-CR approach did not provide a good result in the prefocal zone, preserving only one peak of the three located at the measured field. The after-lobe in the far field was not present for this approximation, even when it included the rim radiation, which could indicate this approach is useless for this particular transducer. However, controlling the value of $n$ in Equation (3) can reduce the amplitude of the focus of the pure SSR condition to make the model fit with the measurements if correlation does not provide an acceptable result. For this paper, correlation was useful, and the value of $m$ and $n$ were easily found by this method, which permitted to find the constants for the modified SSR profile (Bessel-MOD) in Figure 5, with a more accurate result comparable with the measurements.

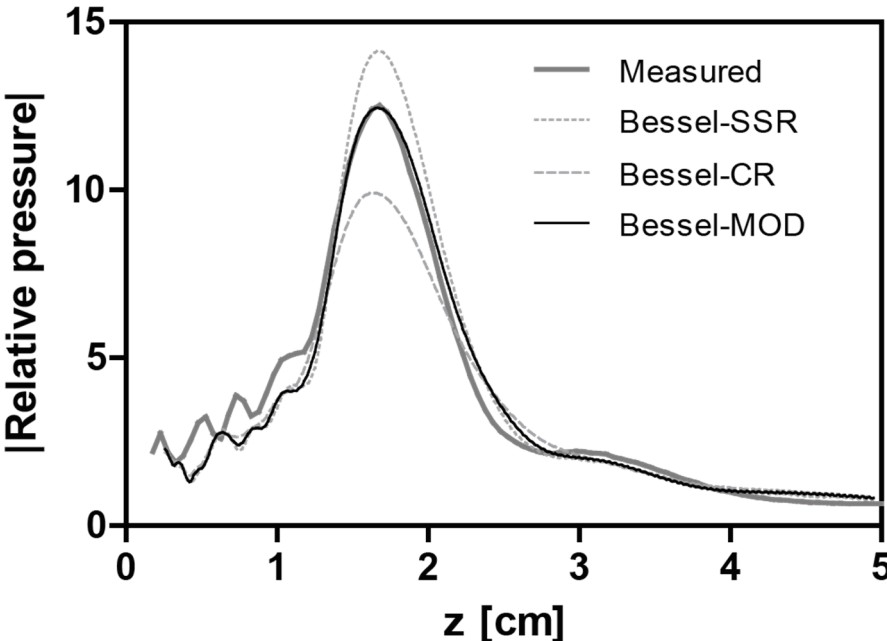

**Figure 5.** The amplitude of the relative pressure on the propagation axis using the Bessel-based nonuniform approximations with SSR and CR conditions on the curved radiating surface. Bessel-MOD is the Bessel-SSR with $n = 0.3$, which emulates correctly the measured field distribution. The pressure is relative to the average emitted pressure. The three models include an 8% of rim radiation, equivalent to the radiation measured at that zone.

The graphs of Figure 6 show the radial profiles at different depths. Important regions were chosen to plot these graphs: Figure 6a is the graph obtained at the first depth measured for the XZ plane; Figure 6b,c is at two equally space depths before the focus; Figure 6d is at the focus; Figure 6e is at the first minimum after the focus; and Figure 6f is at the after-lobe (after the focus). In these figures, the radial profiles at different $z$-distances for the Bessel-MOD are closer to the measurements than the uniform approach, not only in amplitude but also in shape. Other approaches presented in this paper were omitted from this figure for clarity, even if they also would have shown an improvement compared with the uniform approach. For instance, the pure SSR produced in the prefocal zone a profile like the Bessel-MOD without the peaks (a smoother profile), while after the focus, the SSR profiles were the same as those presented for the Bessel-MOD in Figure 6e,f.

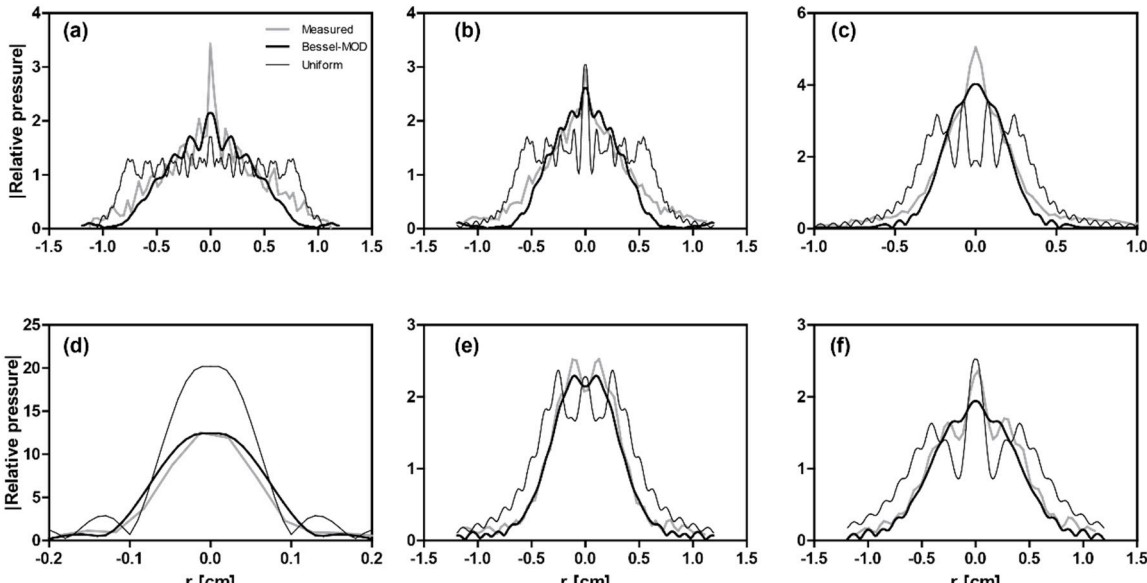

**Figure 6.** The radial relative pressure at different depths of the measured acoustic field compared with the SCUR distribution and the Bessel-MOD distribution, i.e., Bessel-SSR with $n = 0.3$ in Equation (3). Other analyzed distributions did not fully match the measured profile, and they were not included in these graphs for clarity, namely, SSR, CR, and Bessel-CR. The depths are (**a**) 0.2 cm, (**b**) 0.5 cm, (**c**) 1.0 cm, (**d**) 1.7 cm (focus); (**e**) 2.7 cm (down peak); and (**f**) 3.1 cm (lobe after focus). All the graphs have the same type of lines as in (**a**) and the same axis labels. The pressure is relative to the average pressure of Figure 3 of each distribution.

In Figure 7, the full acoustic fields of the more representative models of this work are shown. The Bessel-based approaches with modified SSR (SSR with $n = 0.3$) and CR components are presented in Figure 7c,d, respectively, in order to be compared with the measured field of Figure 7a and the most used classical SCUR distribution of Figure 7b. This ideal SCUR distribution is composed of a very characteristic diffraction pattern before the focus, which is not present in the measured profile. The Bessel-MOD proposal provides a better result in the prefocal zone, with a similar diffraction pattern to the measured field. The Bessel-CR still has a similar diffraction pattern but with a narrower acoustic field and larger focus size, probably because of the reduced effective radiating area [41]. The focus locations in the Bessel-based models were more in agreement with the measured field than the SCUR; this was probably because of the reduction of the effective radiating areas, which were determined from each data set (models and measurements) to normalize the average emitted radiation that was the reference of the relative pressure used in the field comparisons.

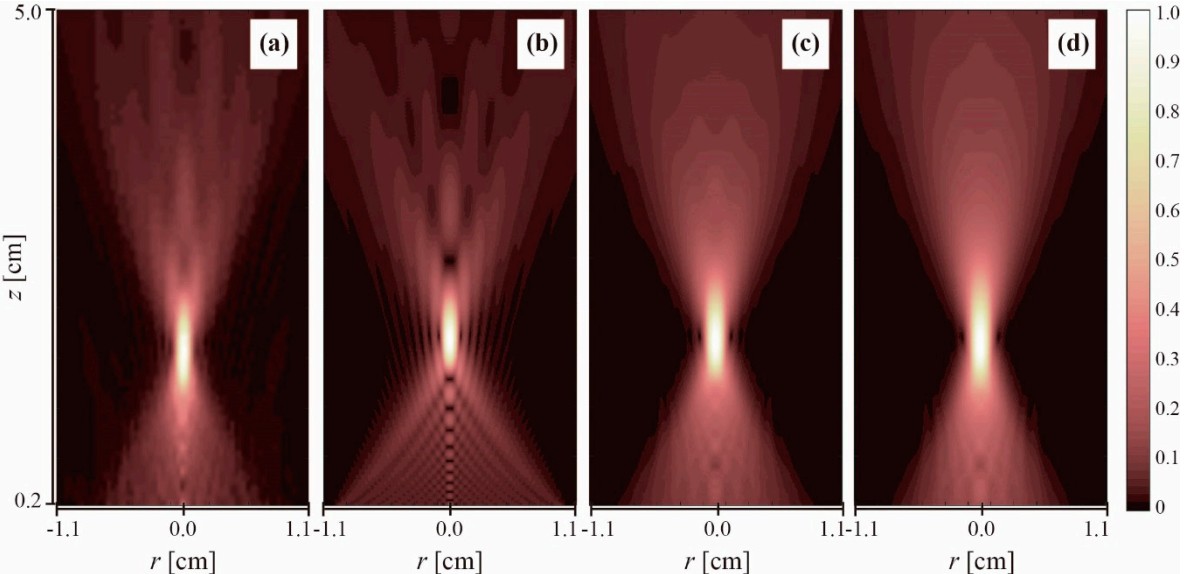

**Figure 7.** The normalized acoustic field of the focused transducer at a central longitudinal plane: the transducer would be at the bottom of each figure. The linear color scale: white = 1 and black = 0. (**a**) The measured acoustic field; (**b**) the modeled field using SCUR; (**c**) the modeled field using a modified Bessel-SSR ($m = 1$, $n = 0.3$); and (**d**) the modeled field using Bessel-CR ($m = 1$, $n = 1$).

## 4. Discussion

A proposal to effectively model the acoustic field of focused transducers has been presented. This was based on another previously proposed nonuniform distribution for planar radiators named Bessel-based nonuniform radiation distributions (shown in Figure 3). In this work, two other more general proposals are also used as comparison: the simply supported radiator (SSR) and clamped radiator (CR). The three proposals have shown that can be used to generate acoustic fields in good agreement with the field produced by focused radiators. The distributions were set in a curved boundary, and the results were obtained with a FEM commercial software. Only low-power linear behavior was considered in this work, but the main rationale presented here could be applied for high-power nonlinear measurements. The emission was considered to be purely harmonic operating at the transducer's nominal frequency of 2 MHz; the measured transducer emission bandwidth was 100 kHz, which represents about 5% of the central operation frequency. Because of that narrow bandwidth, the effect of this parameter in the acoustic field was considered negligible. The use of an analytical expression in the radiating boundary instead of the raw measurements, for acoustic field processing, would open the possibility not only to reduce computation but also to find a simpler way to calculate analytically the acoustic field in a closed form under certain conditions, for instance using the Hankel transform [42] or the direct integration of the Rayleigh integral [22,24]. However, this process is beyond the scope of this paper.

Figure 4 shows the axial acoustic fields for the SCUR approach and the two classical nonuniform distributions SSR and CR already proposed for planar radiators. It can be noticed that the latter produce good results in the overall field amplitude when they are set in this curved radiating boundary but with poor agreement in the prefocal zone. At this region, the diffraction effects are quite different between the measured profiles in both the SSR and CR models. Actually, the diffraction profile of pure SSR and CR fields are similar to the SCUR field but with a reduced amplitude and an "offset". This indicates that reducing the emitted field amplitude at the edges of the radiator provokes also a smooth overall acoustic field. The SSR and CR fields are closer in amplitude to the measured field, and this could occur because the piezoelectric plate of the transducer is fixed at the edges, which produces an emitted radiation closer to a SSR/CR distribution than a SCUR, as seen in the measured field in Figure 3. This is also noticeable in the focus locations, which were closer to the measurements in the

SSR and CR distributions than the SCUR; this was improved probably because of the reduction of the effective radiating area in the nonuniform distributions [15]

Although these classical nonuniform distributions are closer in amplitude to the measured field, their results are not fully satisfactory. Then, it is proposed the use of Bessel-based distributions is closer to the real vibration distribution produced in concave transducers. The results of combining a Bessel with SSR and CR edge conditions are shown in Figure 5. From those results, it can be noticed that the prefocal zone of the acoustic field is better modeled after including the Bessel behavior in the radiation distribution. The peak-distribution in the modeled prefocal zone using Bessel-SSR has the same number of peaks as the measured field and almost at the same position. The focus amplitude was improved when adapting the equation with $n = 0.3$, for the Bessel-MOD, which produced a field more congruent with the measurements. The focus locations of these representations were also more in agreement with the measurements than the classical models presented before, probably due to the reduction of the effective radiating areas. In the post-focus zone, the model can also produce the after-lobe observed in the measurements but with a slightly lower amplitude. This after-lobe was only present in the Bessel-SSR and Bessel-MOD simulations with 8% of rim radiation, which was based on the measurements (see rim radiation in Figure 3). The Bessel-CR condition produced a similar profile at the prefocal zone but with less pronounced peaks. In this case, the after-lobe in the post-focal zone was practically unnoticeable, even with the inclusion of the rim radiation, probably because of the total dominance of the acoustic intensities coming from the main radiating surface.

Figure 6 shows the radial distributions at different depths of the three more significant cases for our comparison purposes, namely, the SCUR, the modified Bessel-SSR (Bessel-MOD), and the measurements. Before the focus, in Figure 6a–c, the radial profile using the Bessel-MOD follows adequately the measured field with some variations in amplitude in the central peak. In spite of the complexity of adequately simulating this region, the Bessel-MOD approach correctly produces acceptable results. A similar agreement was obtained in other less complicated regions as the focus and after the focus in both amplitude and field width. In the former, shown in Figure 6d, the Bessel-MOD produced almost the same amplitude as the measurements, with a 0.4% relative error; the focus with the SCUR is 40% larger than the measured one. After the focus (Figure 6e,f), our model still has a similar tendency as the measured field, with a good match of the graphs in Figure 6e and little variation in the central peak in Figure 6f. From these graphs, the SCUR does not represent an adequate model for this kind of radiators, i.e., those with nonuniform Bessel-based vibration patterns.

The acoustic fields presented in Figure 7 show the differences among each approach in a detailed manner using a linear color scale. The prefocal zones of both Bessel approaches are similar but with a larger focus for the Bessel-CR approach. Focus locations were also improved using our proposed models. The SCUR approach differs at practically any region from the measured field. For this transducer, the Bessel-MOD presents better results than the other nonuniform distributions studied here, with a clear good agreement in most of the graphs shown in this paper. However, this does not mean that the applicability of this model is universal for curved radiators. Some transducers could still behave as ideal pistons, simply supported curved disks, or clamped curved disks that could require the use of any of the other approaches (SCUR, SSR, CR, and Bessel-CR). The use of any of these models should depend on the real radiation profile measured very close to the radiating surface. This paper has presented a new modeling approach to effectively simulate the acoustic field of focused transducers that can be adapted to most of the devices used in different medical applications. Eventually, using the proposed functions as radiation distributions of this kind of transducers could potentially permit to find more analytical solutions of this kind of radiators that could better match the measurements.

## 5. Conclusions

In this paper, we have presented four proposals of nonuniform vibration distributions on the radiating surface of a curved transducer to obtain more realistic simulated acoustic fields very

congruent with field measurements. From the results here shown, it was possible to conclude that these models provide better approximations of the vibration distribution capable of producing accurate representations of acoustic field for focused applications. When using Bessel-based functions for the vibration of the curved radiator, the prefocal zone of the transducer was correctly simulated. For the post-focal zone, our Bessel proposal had to include the radiation coming from the transducer rim, which allowed the incorporation of the central "after-lobe" observed in the measured field. In the other proposed approaches, the amplitude of the focus significantly varied with respect to the measurements. This happens possibly because of the differences in the proportion of the emitted average radiation used for determining the relative pressure and because of the effective radiating area in each condition, which is a parameter rarely considered for this kind of transducers [15].

Having a model to correctly simulate the acoustic field in the prefocal zone for focused radiators is a very important improvement in the field. This will permit to increase the accuracy in therapy planning, to improve the prediction of thermal increments in tissues outside the focus, to produce better thermal models in hyperthermia, and in general, to have a better dose control. The use of more realistic but still simple models of acoustic field for focused radiators will help to control the undesired effects out of the treatment zone, i.e., before and after the focus, and to easily incorporate this model proposal to therapy planning. In this work, it was proven that our models represent better alternatives for focused radiators than the widely used ideal uniform approaches.

**Author Contributions:** Conceptualization, M.I.G. and A.R.; data curation, M.I.G.; formal analysis, M.I.G.; funding acquisition, M.I.G., A.R., A.V., and L.L.; investigation, M.I.G.; methodology, M.I.G. and A.R.; project administration, M.I.G., A.V., and L.L.; resources, M.I.G., J.G., A.V., and L.L.; software, M.I.G.; supervision, M.I.G., A.R., J.G., A.V., and L.L.; validation, M.I.G.; visualization, M.I.G.; writing—original draft, M.I.G.; writing—review and editing, M.I.G. and A.R.

**Funding:** This research was funded by CONACyT, grant number 257966; CSIC, grant number COOPB20166; ERAnet-EMHE CSIC, grant number 200022; Spanish P.N RETOS, grant number DPI2017-90147-R; and CYTED-Ditecrod Network Ref. 218RT0545.

**Acknowledgments:** The authors would like to thank Rubén Pérez Valladares for his technical support during the acoustic field measurements.

**Conflicts of Interest:** The authors declare no conflict of interest.

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
