# Peer review of "Nonuniform Bessel-Based Radiation Distributions on A Spherically Curved Boundary for Modeling the Acoustic Field of Focused Ultrasound Transducers"

_applsci, doi:10.3390/app9050911_

Reviewer 1 Report

The paper proposed a modeling technique for focused beam fields computation and compared the simulation results with experiments.

I have some comments on this paper and would like the authors to answer and properly reflect in this paper.

1. Section 2.1 describes the experimental details in a complicated way. The authors need to add a figure or figures that illustrate(s) the experimental setup, equipments, geometry, and so on.

2. The authors state that use of a focused beam is to provide high intensity ultrasonic fields. In this case, the input to a focused transducer is usually very high to generate a finite amplitude wave that will be able to cause nonlinear ultrasonic waves (higher harmonic waves) in the medium. However, the authors modeled the problem using a linear acoustics theory. The authors need to make necessary comments on this in the paper.

3. In Introduction, the authors present some existing modeling techniques for radiation beam from a concave surface in the literature. The authors also talked about the reason why the modeling results for spherically focused transducers do not agree well with the experimental results. As far as the reviewer knows, another possible reason for this discrepancy is that the actual focal length and diameter are not the same as the nominal focal length and diameter of a focused transducer that the manufacturer provides. In this case, it is necessary to calibrate the focused transducer first to determine the actual focal length and diameter to be used in the modeling. I would like the authors to look at references on this and to add them in the introduction and in the reference.

Calibration of focused circular transducers using a multi-Gaussian beam model. Applied Acoustics, 2018, 133, 182-185.

Calibration of focused ultrasonic transducers and absolute measurements of fluid nonlinearity with diffraction and attenuation corrections. The Journal of Acoustic Society of America, 2017, 142, 984-990.

Author Response

Response to reviewer 1

Comments:

The paper proposed a modeling technique for focused beam fields computation and compared the simulation results with experiments.

I have some comments on this paper and would like the authors to answer and properly reflect in this paper.

1. Section 2.1 describes the experimental details in a complicated way. The authors need to add a figure or figures that illustrate(s) the experimental setup, equipments, geometry, and so on.

R1: A new figure was included as the reviewer suggested. This figure includes the two setups for curvature and acoustic field measurements. We hope this helps clarify the experimental details.

2. The authors state that use of a focused beam is to provide high intensity ultrasonic fields. In this case, the input to a focused transducer is usually very high to generate a finite amplitude wave that will be able to cause nonlinear ultrasonic waves (higher harmonic waves) in the medium. However, the authors modeled the problem using a linear acoustics theory. The authors need to make necessary comments on this in the paper.

R2: The reviewer’s concern is correct and appropriate. We cited in the introduction some works about HIFU that could produce confusion about the objective of our model and future work. Although the transducer was designed for high intensity applications, it was characterized in low intensity regime; our final application also will be in those low intensity levels. We included new statements in third paragraph of introduction to clarify that this procedure is only valid for linear propagation and low intensity focused ultrasound.

3. In Introduction, the authors present some existing modeling techniques for radiation beam from a concave surface in the literature. The authors also talked about the reason why the modeling results for spherically focused transducers do not agree well with the experimental results. As far as the reviewer knows, another possible reason for this discrepancy is that the actual focal length and diameter are not the same as the nominal focal length and diameter of a focused transducer that the manufacturer provides. In this case, it is necessary to calibrate the focused transducer first to determine the actual focal length and diameter to be used in the modeling. I would like the authors to look at references on this and to add them in the introduction and in the reference.

R3: The reviewer provided two adequate references that permit improve the literature revision in the introduction. Those references were included and commented in the introduction and in other parts of the article. As the reviewer indicates, the real focus length is usually different from the nominal length. We believe this is produced by the transducer construction, which modifies the effective radiating area that is different (in our transducer, smaller) than the nominal area. This conclusion is demonstrated in the article and it is now commented in the discussion for each situation. Authors would like to thank the reviewer for the reminder about commenting this topic.

Reviewer 2 Report

The discussion on the radial profile is very interesting, but it is not the only point:

1) The bandwidth of the frequency spectrum should also be discussed...

2) The use of FEM for propagating wave in a fluid medium is not a good choice.

Author Response

Response to reviewer 2

Comments:

Some assumptions are not clearly assumed:

1) linear propagation in homogeneous medium,

R: This assumption is now explicitly indicated in second paragraph of section 2.2

2) constant velocity,

R: This assumption is now explicitly indicated in second paragraph of section 2.2

3) perfectly flat interface,

R: This assumption is now explicitly indicated in third paragraph of section 2.2, when we referred to the external boundaries of the model. However, the geometry does not have internal interfaces, and wave reflections are only important for the walls. The geometry was large enough to have any remaining wave reflection outside the region of interest, i.e. from z = 0 cm to z = 5 cm the acoustic field. A statement was already included about this.

4) no attenuation nor frequency dispersion.

R: This assumption is now explicitly indicated in second paragraph of section 2.2

5) purely harmonic source,

R: This assumption is now explicitly indicated in second paragraph of section 2.2

The first four assumptions (from 1) to 4)) are commonly admitted, but the fifth one is rarely presented nor discussed.

R: The wave equation already considered the source to be purely harmonic. However, as the reviwer indicated, it was not explicitly discussed. Now, this is not discussed in section.

Questions:

Q1: Frequency spectrum of the source

The question that is not answered is that of the frequency spectrum of the source. The simulated transducer has a center frequency of 2 MHz, but we do not have any information on its effective bandwidth. The effect of the bandwidth can be very strong too.

R: This is an important aspect that is now commented in the discussion. The transducer itself has a narrow bandwidth, mainly because of its construction (basically composed by the curved piezoelectric disk, electrodes, no backing, no matching layers). During acoustic field measurements, the transducer was excited with a narrowband sine signal (bandwidth 135 kHz). The signal was emitted and then measured by the hydrophone; that emitted signal had a bandwidth of 100 kHz. Next, we are showing the measured conductance of the transducer that is related with its nominal bandwidth (Ramos-Fernandez, 1985), and both the excitation signal and the measured signal by the hydrophone (labeled as emitted signal). Both bandwidths are enough narrow (about 5% of the central frequency) to consider negligible that effect in the results.

   (FIGURES in file attached)

Reference

-     A. Ramos-Fernandez, F. Montoya-Vitini, and J. A. Gallego-Juarez, “Automatic System for Dynamic Control of Resonance in High-Power and High Q-Ultrasonic Transducers,” Ultrasonics, vol. 23, no. 4, pp. 151–156, 1985.

Q2: Propagation code

The FEM is a very versatile tool, but it is not the optimum one at all. Some very efficient alternatives are existing an widely developed in the literature:

Discrete Hankel Transform, variable radius sum decomposition, ...

* Some references on this point are summarized in the following paper, which deals with a similar approach to that developed in this paper:

* An other related work is the following one:

http://iopscience.iop.org/article/10.1143/JJAP.46.3077

R: This is an interesting comment. We made a revision of the estate of the art of methods to calculate the acoustic field of different transducers, from planar to focused, axisymmetric and not axisymmetric, annular arrays, etc. Among those methods (some of them used by our group in other publications, with other purposes), we specifically chose the FEM because of the future scope of our work, which is to include more complicated anatomical geometries with non-homogeneous media (biological tissues) and to include other physical phenomena to be coupled with the acoustic field, as the bio-heat equation. However, the references that the reviewer provided were useful to open the panorama presented in the introduction and to be properly discussed when comparing our approach with other resources used previously by other research groups. Those references were included respectively in both sections, introduction and discussion.

Remarks:

Page 4: line 171 to 174

* The boundary 6 contribution to the field is somewhat erroneous and strangely argued. How this 8% of the average can be explained?

R: This aspect is now more clearly explained in the paper. When simulating the Bessel-MOD without rim radiation, the pre-focus pattern did not importantly changed; however, this little variation should have been mentioned in the paper. We noticed that the rim contribution was more important just after the focus (see figure below), where the models without this consideration failed. As the reviewer indicates, the argument in the paper was not correct, since it omitted more other effects of the rim radiation, as also shown in the figure below for pure rim radiation.

About the determination of the 8%, it was based on the measurements, and that value is relative to the average of the pressure radiated by the effective radiation area. The relative pressure calculation was determined using this procedure that was already detailed in (Gutierrez 2012) for planar radiators and works by other groups before. In this paper, we make an attempt to propose a way of determining an effective radiating area for focused transducers, also used by (Zhang 2018) determined in a different manner, which accordingly produces good results in focus amplitude and location comparable with measurements. With this procedure, we avoided the use of normalization for field comparisons.  

(FIGURE in file attached)

Acoustic field on the propagation axis with different radiating conditions. Blue line, Bessel-MOD with rim radiation (that one of the paper). Red line, Bessel-MOD without rim radiation. Green line, only rim radiation (no ultrasound coming from the curved transducer boundary). Black line, same as green line but x10 to see the pattern.

References

-       M. I. Gutierrez, H. Calas, A. Ramos, A. Vera, and L. Leija, “Acoustic Field Modeling for Physiotherapy Ultrasound Applicators by Using Approximated Functions of Measured Non-Uniform Radiation Distributions,” Ultrasonics, vol. 52, no. 6, pp. 767–777, 2012.

-       Zhang, S.; Hu, P.; Li, X.; Jeong, H. Calibration of focused circular transducers using a multi-Gaussian beam model. Appl. Acoust. 2018, 133, 182–185.

* The calculation time for a given processing setup was not given as an example. It would be very interesting to have some values on this point.

R: A representative solution time was included now in section 2.1 first paragraph.

Page 6 : line 227

* “The function f(r) is given by Eq. (3)”: There is no f(r) in Eq.(3)...

R: Thank you for your observation. That text was rewritten to be clearer. The Eq. 3 provides the function f(r) in Eq. 4. Eq. 3 is used to represent the type of edge fixing in Eq. 4. We hope after the addition of the new text, this part does not produce any misunderstanding.

Page 9 :

* Discussion :

The possibility of alternatives for the calculation of the wave propagation in a fluid is mentioned at the end of the paper. Maybe calculation time and resources is not a problem any more ?

“ The advantage of using an equation in the radiating boundary instead of the raw measurements for acoustic field processing, is indeed the simplification of postprocessing and modeling, and the possibility to determine a simpler way to calculate analytically the acoustic field in a closed form, for certain conditions. However, this process is beyond the scope of this paper. ”

R: That section was edited to be more understandable. A more detailed comment about other alternatives for acoustic field calculation was also made using the references the reviewer kindly provided.
